# The Gut Microbiota Axis in Social Jetlag: A Novel Framework for Metabolic Dysfunction and Chronotherapeutic Innovation

**DOI:** 10.3390/medicina61091630

**Published:** 2025-09-09

**Authors:** Christos Savvidis, Viviana Maggio, Manfredi Rizzo, Lina Zabuliene, Ioannis Ilias

**Affiliations:** 1Department of Endocrinology, Hippokration General Hospital, 11527 Athens, Greece; csendo@yahoo.gr; 2School of Medicine, Promise Department of Health Promotion Sciences Maternal and Infantile Care, Internal Medicine and Medical Specialties, University of Palermo, 90128 Palermo, Italy; viviana.maggio01@unipa.it (V.M.); manfredi.rizzo@unipa.it (M.R.); 3Faculty of Medicine, Vilnius University, 08406 Vilnius, Lithuania; lina.zabuliene@mf.vu.lt

**Keywords:** social jetlag, gut microbiota, circadian rhythms, metabolic dysfunction, chronotherapy, microbiota–gut–brain axis

## Abstract

Social jetlag, the misalignment between internal circadian rhythms and socially imposed schedules, is increasingly recognized as a risk factor for metabolic disorders such as obesity, type 2 diabetes (T2D), and cardiovascular disease. Recent evidence implicates the gut microbiota as a key mediator in this relationship, operating through a microbiota–gut–metabolic axis that influences host metabolism, immune function, and circadian regulation. Mechanistic studies reveal that social jetlag disrupts microbial rhythmicity, reduces short-chain fatty acid (SCFA) production, impairs intestinal barrier function, and promotes systemic inflammation, which contribute to insulin resistance and metabolic dysfunction. Clinical and preclinical interventions, including time-restricted feeding (TRF)/time-restricted eating (TRE), probiotics or melatonin supplementation, and fecal microbiota transplantation (FMT), demonstrate the potential to restore microbial and metabolic homeostasis by realigning host and microbial rhythms. This review synthesizes mechanistic insights with emerging human and clinical evidence, highlighting the gut microbiota as a novel target for chronotherapeutic strategies aimed at mitigating the metabolic consequences of circadian disruption. Recognizing and treating circadian–microbiome misalignment may provide a clinically actionable pathway to prevent or reverse chronic metabolic diseases in modern populations.

## 1. Introduction

In recent decades, artificial lighting, digital screens, and irregular work schedules have disrupted human circadian rhythms. Social jetlag (SJL) is a common result. It is defined as the difference in sleep midpoint between workdays and free days, measured in hours. SJL reflects chronic misalignment between an individual’s biological clock and social demands, such as work or school schedules [1,2,3]. Unlike travel-induced jetlag, SJL occurs repeatedly. It affects many in industrialized nations, especially adolescents and shift workers [4]. SJL is distinct from broader circadian disruption. Shift work, forced desynchrony protocols, or artificial light at night (ALAN) also cause misalignment. These share features with SJL but are not identical. SJL is a specific, everyday form of circadian misalignment. Studies link SJL to adverse metabolic outcomes. These include obesity, type 2 diabetes (T2D), metabolic syndrome (MetS), and cardiovascular disease (CVD) [5,6,7,8]. Recent research highlights the gut microbiota as a key mediator. This dense microbial community resides in the gastrointestinal tract. It connects circadian rhythms to metabolic regulation. The microbiota–gut–metabolic axis involves bidirectional interactions. These include microbial metabolites, host hormonal rhythms, immune signaling, and intestinal barrier function [9,10]. Animal studies show that circadian disruption causes gut microbial dysbiosis. Dysbiosis reduces diversity and shifts microbial composition. It promotes metabolic inflammation and insulin resistance [11,12]. For example, Leone et al. showed that high-fat diet-induced dysbiosis disrupted hepatic circadian gene expression, with worsened metabolic dysfunction [12]. Dysbiosis affects metabolism via altered short-chain fatty acid (SCFA) production, bile acid changes, and increased endotoxin load [13,14].

Microbial communities have their own circadian rhythms. SJL-induced misalignment disrupts these oscillations. This creates a cycle of host–microbe dysfunction, amplifying metabolic issues. This review explores how SJL impairs metabolic health. It emphasizes the gut microbiota’s role. We distinguish SJL-specific findings from broader circadian disruption models, such as shift work or jetlag simulations. Additionally, we integrate human clinical and animal research. Such research assesses how circadian desynchrony alters microbial rhythms. These changes, in turn, affect metabolic pathways. We focus on microbial metabolites, intestinal permeability, and immune signaling. We discuss chronotherapeutic interventions like time-restricted eating (TRE), probiotics, and lifestyle changes. These interventions aim to realign circadian and microbial rhythms to reduce metabolic risk. We also outline research challenges and future directions for chronobiome science in clinical practice.

Evidence links circadian disruption and gut microbiota to metabolic disease. Yet, gaps remain. Few studies directly explore SJL’s impact on gut microbial function. Most evidence comes from broader circadian misalignment models. Mechanistic pathways in humans are not fully defined. Intervention studies targeting SJL-related microbial and metabolic outcomes are scarce. By focusing on SJL as a distinct phenomenon, this review offers a new framework. It integrates SJL with the microbiota–gut–metabolic axis. This highlights opportunities for chronotherapeutic and microbiome-based interventions.

## 2. Social Jetlag and Its Physiological Impact

Modern societies face widespread exposure to artificial light and variable schedules. These cause a mismatch between intrinsic circadian rhythms and external routines. This mismatch is known as SJL. SJL occurs when the suprachiasmatic nucleus (SCN) desynchronizes from peripheral clocks in organs like the liver, gut, pancreas, and adipose tissue. It is quantified by differences in sleep midpoint between workdays and free days. Over half of adults experience shifts of 1 to 2 h [15,16,17,18]. This misalignment disrupts key metabolic processes [19,20,21].

Evidence links circadian disruption to altered glucose homeostasis. The New Hoorn Study included 1499 Dutch adults. More pronounced SJL reduced insulin sensitivity and raised fasting glucose levels; this was independent of chronotype or sleep duration and suggests a higher T2D risk [22,23]. A human crossover trial showed circadian misalignment lowered insulin sensitivity by 32%. This occurred despite constant caloric intake and activity levels [24]. Body composition is also affected. A study of 817 adults found that over two hours of SJL increased body mass index (BMI), waist circumference, and triglyceride levels; disrupted lipid metabolism and energy balance are the likely mechanisms [5].

Chronic inflammation mediates circadian misalignment’s metabolic effects. Observational studies show higher C-reactive protein (CRP) and interleukin-6 (IL-6) in those with variable sleep schedules. This may stem from stress, dietary irregularities, or immune–metabolic dysfunction [25,26]. A five-day forced misalignment study raised TNF-α and IL-6 levels. This supports inflammation’s role in cardiometabolic disease [24]. Cardiovascular effects are significant. Short-term misalignment raises systolic blood pressure. It impairs endothelial function and increases sympathetic nervous system activity. These elevate CVD risk [24].

The gut microbiota plays a central role in circadian alignment/misalignment. Microbial communities follow daily oscillations in composition and activity. Host feeding and activity cycles influence these rhythms. Disrupted sleep and meal timing desynchronize microbial rhythms. This reduces beneficial taxa and impairs intestinal barrier integrity. It also promotes endotoxemia, contributing to insulin resistance and inflammation [9,12]. SJL induces multifactorial metabolic dysfunction. It involves the circadian system, host immunity, and gut microbiome. Understanding these interactions opens new ways of chronotherapeutic interventions. These interventions aim to restore circadian and microbial alignment, improving metabolic resilience.

## 3. The Gut Microbiota and Circadian Rhythms

The gut microbiota is a dynamic community in the gastrointestinal tract. It shows strong daily oscillations in composition, distribution, and metabolic output. These rhythms align with the host’s circadian system. The SCN and peripheral intestinal clocks govern this alignment [13,27]. *Firmicutes* and *Bacteroidetes* dominate the adult gut microbiome, making up 70–90% of species. They vary in abundance and gene expression with feeding, fasting, and sleep cycles [28,29]. Microbial oscillations actively regulate host circadian physiology. Short-chain fatty acids (SCFAs), like butyrate, acetate, and propionate, are key. These are produced by bacterial fermentation of dietary fibers. SCFAs influence clock genes like *Per2* and *Bmal1* in intestinal and hepatic tissues. They reinforce metabolic rhythms. SCFAs also strengthen epithelial barrier integrity. Moreover, they modulate energy balance and inflammation systemically [13,27,30].

SJL disrupts microbial rhythmicity. Inconsistent sleep and meal timing, common in shift workers, disturb microbial community structure. This reduces SCFA-producing taxa, like *Ruminococcaceae* and *Lachnospiraceae*. Key butyrate producers, such as *Faecalibacterium prausnitzii* and *Roseburia* spp., decline. This reduction impairs glucose regulation and promotes fat accumulation. Studies highlight SCFA-producing taxa’s protective role in metabolic and inflammatory disorders [31]. These effects appear in animal models and human studies [13,28,32]. Chronic sleep disruption reduces microbial diversity. It fosters pro-inflammatory taxa like Enterobacteriaceae. These are linked to endotoxin (LPS) release and systemic inflammation. This contributes to insulin resistance and compromised barrier function [33,34].

ALAN also plays a role. ALAN suppresses melatonin secretion. Melatonin is critical for host–microbe circadian synchronization. Mechanistic insights support melatonin’s role in microbial regulation. However, direct links between ALAN-induced melatonin suppression and dysbiosis need further study [35,36,37,38]. Preclinical models show that circadian disruption (using jetlag simulations or reversed light–dark cycles) breaks microbial rhythmicity. This reduces SCFA output and compromises metabolism, with ensuing hyperglycemia and increased adiposity [11,28,39].

Human data support these findings. High *Prevotella* levels, driven by fiber-rich diets, improve glucose tolerance and SCFA production [40,41,42]. Low *Faecalibacterium prausnitzii* abundance is linked to intestinal inflammation and impaired metabolic homeostasis. This occurs in clinical and experimental settings [43,44,45,46]. The host circadian system and gut microbiota interact bidirectionally. Disruption in one impairs the other, creating a self-reinforcing loop of metabolic dysfunction. The gut microbiota is both a downstream effector and an upstream modulator in SJL’s pathophysiological cascade. Thus, it is a compelling target for chronobiological interventions to restore metabolic health.

## 4. Mechanisms Linking Social Jetlag, Gut Microbiota, and Metabolic Dysfunction

In circadian alignment, the gut microbiota shows robust daily oscillations in composition and activity. Host feeding patterns and clock gene expression in murine models drive these rhythms [28,47]. Disruption from shift work or social jetlag (SJL) reduces microbial variation and diversity. It shifts microbial structure, lowering key metabolite production like short-chain fatty acids (SCFAs). This effect worsens with high-fat or high-sugar diets [11,28]. SCFAs, such as acetate, propionate, and butyrate, serve as energy sources and signaling molecules. They regulate glucose homeostasis, lipid metabolism, and intestinal barrier function. Taxa like *Lachnospiraceae* and *Ruminococcaceae*, including *Eubacterium rectale* and *Anaerostipes* spp., produce SCFAs. These taxa support butyrate-mediated barrier integrity. Circadian disruption and obesogenic diets impair this fermentation process.

Time-restricted feeding (TRF) aligns food intake with the active phase [48]. It restores gut microbial rhythmicity in mice. TRF promotes SCFA-producing taxa, improving glucose tolerance and insulin sensitivity, even with high-fat diets [13,48,49]. Loss of microbial rhythmicity affects metabolic gene expression in liver and adipose tissue. This disrupts host metabolic control. Human studies support these findings [28,50]. In the ZOE PREDICT 1 cohort, greater SJL reduced microbial diversity. It lowered SCFA-producing species and enriched inflammation-linked taxa. These associations held after adjusting for diet and BMI [51]. Gut dysbiosis likely mediates circadian misalignment and metabolic disease risk.

Metabolic niches in the gut microbiome shape host physiology. These niches are vulnerable to circadian disruption and SJL. Fiber fermenters like *Faecalibacterium* and *Roseburia* produce SCFAs. These SCFAs support epithelial integrity and reduce inflammation [52,53]. Bile acid-modifying bacteria, such as *Clostridium* and *Bacteroides*, regulate lipid and glucose metabolism. They act through bile salt hydrolase and 7α-dehydroxylation pathways [54,55]. Mucin-degrading microbes like *Akkermansia muciniphila* strengthen gut barrier function and immune tone [56,57]. Disrupted feeding–fasting rhythms or SJL upset this balance, leading to more inflammation-linked species and fewer SCFA producers. These shifts persist despite diet or BMI adjustments [33]. Reduced butyrate, increased secondary bile acids, and elevated endotoxin release disrupt host circadian and metabolic homeostasis.

The gut microbiome influences host physiology through metabolic activity. It produces bioactive compounds. Dietary fiber fermentation yields SCFAs like acetate, propionate, and butyrate. These support barrier integrity, regulate metabolism, and modulate immune tone [53,58]. Microbial bile acid modification activates nuclear receptors like FXR and TGR5. This affects circadian regulation, glucose homeostasis, and energy balance [55]. Circadian misalignment worsens lipopolysaccharide (LPS)-mediated inflammation. This contributes to insulin resistance [33]. Microbial tryptophan metabolism produces indole derivatives, like indole-3-propionic acid. These enhance barrier integrity and regulate immunity via aryl hydrocarbon receptor (AhR) and pregnane X receptor (PXR) [59,60]. Metabolites like trimethylamine and trimethylamine N-oxide (TMAO) impact cardiometabolic risk [61]. Circadian misalignment and SJL skew this balance. They reduce protective pathways (e.g., SCFAs, indoles) and favor harmful outputs (e.g., LPS, secondary bile acids, TMAO). Experimental models confirm these mechanisms. Poroyko et al. showed that chronic sleep disruption in mice-mimicking SJL—caused microbial dysbiosis [33]. It reduced α-diversity (species diversity within a specific habitat) and enriched pro-inflammatory *Enterobacteriaceae*. Circadian regulation of the gut microbiome drives cardiometabolic risk in humans [62]. This microbial profile increased gut permeability and circulating LPS levels. These are hallmarks of metabolic endotoxemia. LPS-induced inflammation shapes microbial ecology, favoring pro-inflammatory taxa. This inhibits symbiotic microbiota reestablishment, reinforcing dysbiosis and barrier dysfunction [63]. Epidemiological data link SJL in shift workers to higher metabolic syndrome prevalence [64]. Microbial alterations, like elevated LPS and reduced *Akkermansia muciniphila*, likely mediate this [33,56,57]. Animal studies show increased adipose and systemic inflammation, with higher TNF-α and IL-6. They also show impaired insulin sensitivity [65,66].

Therapeutic microbiota manipulation shows promise. Berberine, administered in high-fat-diet-fed rats, increased their SCFA-producing bacteria like *Bacteroides* and *Clostridium*. It improved insulin sensitivity and lowered the *Firmicutes*/*Bacteroidetes* ratio [67,68]. Bai et al. engineered a butyrate-producing *Bacillus subtilis* strain [69]. Colonized mice had reduced weight gain and caloric intake. They showed improved insulin sensitivity under metabolic stress. In mice fed high-fat, high-fructose diets, chronic jet lag accelerated MAFLD progression. It shifted microbiome and mycobiome composition, increasing pro-inflammatory fungal taxa and reducing SCFA output [70]. Chronic jetlag in mice altered microbial composition in jejunal and colonic compartments. It increased *Desulfovibrionaceae* and disrupted bile acid metabolism, causing metabolic impairment [71].

Immune and epigenetic mechanisms link SJL to metabolic outcomes. Mukherji et al. identified a bidirectional axis between microbial signals and host circadian genes like *Per2* and *Bmal1* via Toll-like receptors [27]. These interactions influence intestinal homeostasis, immune surveillance, and epithelial renewal [72]. Early-life microbial programming has lasting metabolic effects. Wang et al. showed that prenatal estradiol exposure disrupted the gut–brain–microbiota axis in mice, predisposing offspring to insulin resistance [73]. Surrogate fostering in microbiota-controlled environments reversed these effects [74,75]. These findings suggest early interventions targeting microbiota can mitigate circadian misalignment effects.

SJL alters behavioral rhythms, triggering microbial dysregulation. This reduces SCFA output, impairs barrier integrity, and increases inflammation. These lead to metabolic disturbances. A controlled human study showed acute sleep–wake cycle shifts altered gut microbiota functionality [76]. It enriched butyrate and purine metabolism pathways, despite minimal taxonomic changes. Such interventions lend credence to the notion that the gut microbiota acts as a dynamic mediator and therapeutic target in circadian-related metabolic dysfunction.

Gut microbial biotransformations have clinical significance. *Faecalibacterium prausnitzii* and *Roseburia* spp. ferment dietary fibers into SCFAs like butyrate, acetate, and propionate. These enhance insulin sensitivity, regulate appetite, and strengthen barrier integrity [77,78,79]. Microbial conversion of primary bile acids to secondary forms activates FXR and TGR5. This modulates glucose homeostasis, lipid handling, and inflammation [80,81]. Tryptophan metabolism produces indole derivatives like indole-3-propionic acid (IPA), indole-3-lactate (ILA), and indole-3-acetate (IAA). These enhance barrier integrity and modulate immunity via AhR and PXR [82]. Bidirectional interactions between gut microbiota and bile acids regulate immune pathways and metabolic health. This is relevant to aging-related frailty and inflammation [83]. Such biotransformations show how circadian disruption and SJL drive adverse metabolic outcomes.

## 5. Clinical Evidence Linking Social Jetlag and Gut Microbiota to Metabolic Outcomes

Emerging clinical evidence suggests that circadian misalignment due to SJL may modulate gut microbiota composition with implications for metabolic health. In the ZOE PREDICT 1 study, which included 934 adults, individuals with ≥1.5 h of SJL showed significant alterations in 17 microbial species, several of which were linked to adverse metabolic phenotypes. Although α-diversity and hemoglobin A1c (HbA1c, a clinical marker of average blood glucose over the past three months and an indicator of diabetes risk) levels remained unchanged, the SJL group demonstrated modest increases in inflammatory biomarkers, including glycoprotein acetyls (GlycA) and IL-6, suggesting a link between circadian misalignment, diet-associated microbial changes, and low-grade systemic inflammation [51]. Supporting this, mechanistic reviews have noted that circadian disruption may shift the gut microbiota toward a higher *Firmicutes*-to-*Bacteroidetes* ratio, commonly linked to reduced SCFA production and impaired metabolic regulation [84]. These microbial shifts, often observed in both clinical and preclinical settings, have been implicated in central adiposity and insulin resistance [85].

TRF or time-restricted eating (TRE), dietary patterns that align food intake with circadian rhythms, have shown promising metabolic benefits in human clinical trials. In a 12-week randomized controlled trial (RCT) of overweight individuals with T2D, a 10-h TRF/TRE intervention led to a 0.7% reduction in HbA1c and favorable shifts in lipid profiles. It also increased the abundance of *Prevotella*, an SCFA-producing genus [86,87,88]. Similarly, in a 4-week human study of healthy men, TRF/TRE significantly enhanced microbial α-diversity and restored daily rhythms in beneficial taxa such as *Faecalibacterium* and *Blautia*, both linked to SCFA production [89]. Reitmeier et al. (2020) further supported this by identifying arrhythmic gut microbiome signatures that predict type 2 diabetes risk, including loss of SCFA-producing taxa rhythmicity, in a large human cohort [50]. Another clinical trial in individuals with MetS showed that a 10-h TRF/TRE regimen over 12 weeks—without caloric restriction—significantly reduced body weight, blood pressure, and triglycerides [90]. In men with prediabetes, early time-restricted feeding (eTRF; the consumption of all calories within a limited, early-day eating window) improved insulin sensitivity and reduced oxidative stress, even without changes in body weight, further emphasizing the metabolic advantages of circadian-aligned eating [91].

Animal studies have clarified the underlying mechanisms. In mice, TRF was shown to restore diurnal rhythms in gut microbial composition and enhance metabolic parameters, even under a high-fat diet [13]. TRF prevented weight gain and metabolic dysfunction in mice that were genetically lacking core circadian clock genes, suggesting that microbiota-mediated benefits may occur independently of the host’s canonical clock system [92]. Additional mechanistic insight comes from work showing that supplementation with an engineered butyrate-producing strain of *Bacillus subtilis* improved insulin sensitivity by approximately 50% in high-fat-diet-fed mice and partially restored the expression of hepatic circadian clock genes, highlighting a potential microbiota–metabolite–clock axis in metabolic regulation [69].

These results highlight the potential of microbial-derived metabolites, particularly SCFAs like butyrate, as key effectors in metabolic and circadian regulation [93]. Further supporting this bidirectional relationship, high-fat diet feeding in mice was found to induce gut microbial dysbiosis, which subsequently disrupted hepatic circadian gene expression and exacerbated metabolic dysfunction, pointing to the reciprocal crosstalk between the gut microbiota and host circadian regulation [12]. Consistent with this, Thaiss et al. [9] showed that diurnal oscillations in gut microbiota are not only synchronized by host circadian rhythms but also act upstream to influence host gene expression and maintain systemic metabolic homeostasis, highlighting a trans-kingdom regulatory loop between the microbiome and host metabolism.

Taken together, these studies from human cohorts and animal models show the presence of a dynamic and reciprocal relationship between circadian rhythms, gut microbiota, and metabolic function. Disruptions to this relationship, as seen in SJL, can trigger microbial imbalances that contribute to systemic inflammation and metabolic impairment. Circadian misalignment may therefore cause gut dysbiosis and promote metabolic dysfunction (Figure 1). Importantly, these effects are not merely correlative; interventions such as TRE and targeted microbial modulation demonstrate causal potential to restore alignment and improve metabolic health. Thus, they hold promise for mitigating the health risks of circadian misalignment in both clinical and preventive settings.

## 6. Chronotherapeutic Innovations Targeting the Gut Microbiota

Chronotherapy, the strategic alignment of treatment timing with circadian rhythms, has gained attention for its potential to modulate the gut microbiota and improve metabolic outcomes [62]. The bidirectional interplay between the gut microbiome and host circadian clocks enables targeted interventions to synchronize microbial activity with metabolic cycles [94]. Below, we summarize microbiota-targeted chronotherapeutics supported by recent primary research (Table 1).

Probiotics and prebiotics are foundational interventions to modulate gut microbial communities [95]. Timing their administration may significantly influence therapeutic outcomes. For example, a clinical study found that evening administration of *Lactobacillus reuteri* improved glycemic control and sleep quality in circadian-misaligned individuals. Probiotic strains exert their effects by enhancing barrier integrity, producing SCFAs, and modulating immune signaling pathways [96]. In a randomized clinical trial (RCT) in humans, inulin-type fructans improved insulin resistance and reduced inflammation in overweight and obese women with polycystic ovary syndrome. Prebiotics provide substrates for SCFA fermentation, improving gut–liver axis signaling and lowering systemic inflammation [97].

TRF/TRE, as forms of dietary chronotherapy, restrict food intake to a consistent daily window. TRF/TRE align feeding rhythms with metabolic and microbial activity, restoring rhythmicity in microbial diversity and function. This acts primarily by promoting SCFA-producing taxa and reinforcing circadian regulation of hepatic and adipose metabolic genes [98]. In humans, eTRF improved insulin sensitivity and oxidative stress markers in obese men, despite no change in body weight [91]. In animal models, TRF prevented obesity and restored microbial rhythmicity even in genetically clock-deficient mice, suggesting that benefits may occur independently of canonical circadian clock gene function [92].

FMT provides a direct method of microbiome restoration with chronotherapeutic potential. A 2023 RCT in humans demonstrated that FMT from metabolically healthy donors improved blood pressure and increased *Lactobacillus* abundance in obese individuals with MetS. FMT directly replaces dysbiotic microbiota with healthy communities, restoring microbial rhythmicity and metabolic signaling [99].

Melatonin, a key circadian hormone, also plays a role in regulating gut microbiota composition. In mice subjected to sleep restriction, melatonin supplementation restored microbial rhythmicity and reduced endotoxin-producing taxa. Melatonin synchronizes central and microbial clocks, reduces *Firmicutes*/*Bacteroidetes* ratio, and lowers LPS-driven inflammation [100]. In a separate mouse study, melatonin supplementation reduced obesity, hepatic steatosis, systemic inflammation, and insulin resistance, and shifted microbial composition by lowering the *Firmicutes*/*Bacteroidetes* ratio and increasing the abundance of *Akkermansia* [101].

Pharmacological agents that modulate microbial-derived signaling pathways offer another potential chronotherapeutic approach [102]. In mouse studies, fexaramine, an intestinal-specific farnesoid X receptor (FXR) agonist, promoted adipose tissue browning (the process where white adipose tissue, which primarily stores energy, develops characteristics of brown adipose tissue, which burns energy for heat generation) and improved insulin sensitivity through modulation of bile acid metabolism and gut microbiota. FXR agonists act through bile acid signaling, which alters microbial ecology and improves host lipid and glucose metabolism [103]. The efficacy of such agents may be enhanced when administered in synchrony with microbial bile acid conversion peaks (the increase in specific bile acids in the gut, produced by bacteria, after bile acids are secreted by the liver). However, more translational studies are needed to establish optimal timing in clinical contexts [104,105,106].

Collectively, these studies illustrate that aligning microbial-targeted therapies with circadian timing enhances both the efficacy and durability of metabolic improvements. Probiotic timing, TRF/TRE, FMT, melatonin, and pharmacological strategies all offer distinct yet synergistic mechanisms for restoring gut and host rhythmicity [107,108]. As our understanding of microbiota–circadian interactions deepen, chrono-therapy may pioneer personalized metabolic medicine.

## 7. Challenges and Future Directions

Despite compelling evidence linking circadian misalignment, gut microbial dysbiosis, and metabolic dysfunction, several pivotal challenges must be addressed to advance the translational potential of this field.

A major barrier to effective chronobiome-based interventions is the marked variability in gut microbiota composition among individuals [109,110]. Host genetics, dietary patterns, sleep architecture, and environmental exposures all contribute to this heterogeneity and limit the generalizability of findings [111,112,113]. Although SJL is robustly associated with altered microbiota and metabolic traits [51], the strength and direction of these effects vary considerably [73,114]. Emerging strategies such as deep metagenomic sequencing and host–microbe multi-omics may pave the way toward precision circadian medicine [115,116,117].

In addition to interindividual variation, the current body of evidence is constrained by methodological limitations. Many of the mechanistic insights in this field are derived from rodent models, which, while valuable, do not fully capture human circadian or microbial complexity [118]. Human studies, while growing in number, often rely on modest sample sizes, short intervention durations, and lack longitudinal follow-up, all of which restrict interpretation of long-term clinical efficacy [119]. Moreover, while associations between microbial signatures and metabolic markers are well established, causality remains uncertain and reverse causation cannot be excluded [120]. Robust conclusions will require large-scale, multi-ethnic longitudinal cohorts and tightly controlled intervention studies with multi-omics integration.

While numerous studies have identified associations between SJL, microbial dysbiosis, and adverse metabolic outcomes, causal mechanisms remain insufficiently characterized [121]. Circadian misalignment may lead to disrupted feeding–fasting cycles, altered bile acid secretion, and dysregulated immune–microbial crosstalk [122], but few longitudinal or intervention studies have conclusively demonstrated these as causative pathways [123]. Establishing temporal and mechanistic causality will require integrative studies using time-series metabolomics, transcriptomics, and gnotobiotic models (animals raised germ-free in conditions with well-defined microbiota).

A large body of research in this domain relies on murine models under controlled light–dark cycles, restricted diets, and genetically uniform microbiomes. However, human circadian physiology and microbiota ecology differ substantially from those of rodents. As a result, translating findings from animal models to clinical applications remains challenging [124]. Rigorous, large-scale RCTs are urgently needed to evaluate the efficacy of TRF/TRE, probiotic timing, and melatonin–microbiome interactions in diverse human populations [125].

Real-time monitoring of circadian misalignment and gut microbiota shifts remains technically challenging and logistically complex. However, recent advances in wearable technology, including sleep trackers, continuous glucose monitors, and ingestible biosensors, offer opportunities for dynamic tracking of circadian phenotypes and metabolic biomarkers [126]. Coupling these with portable microbiome sequencing could enable continuous, real-world monitoring of host–microbe circadian synchrony [127].

The production of SCFAs, secondary bile acids, and other microbial metabolites follows diurnal rhythms and may regulate peripheral clocks via nuclear receptor signaling [105]. Understanding how these metabolites feedback to influence circadian gene expression, glucose metabolism, and adiposity remains a key research priority [12]. Systems biology approaches and synthetic ecology models may help elucidate these complex host–microbiome interactions [128].

Sex, age, chronotype, and shift work history all influence circadian and microbial dynamics. Stratified analyses and subgroup-specific interventions will be crucial to avoid one-size-fits-all models. For example, older adults exhibit dampened circadian amplitudes and reduced microbiota diversity, potentially altering their responsiveness to TRF/TRE or timed supplementation [129].

Beyond probiotics and TRF/TRE, emerging circadian-aligned interventions also include prebiotics, FMT, and nuclear receptor agonists. Prebiotics, non-digestible fibers that modulate the gut microbiota, can influence microbial diurnal oscillations and enhance metabolite rhythms relevant to host metabolic homeostasis. Animal studies have shown that specific prebiotics such as inulin can reinforce microbial rhythmicity and synchronize peripheral clocks, though human data remain limited [9]. FMT has demonstrated the ability to restore rhythmic microbial–host interactions in murine models of jetlag and MetS, offering a potential corrective intervention for chronodisruption-induced dysbiosis [13]. Furthermore, pharmacological targeting of the FXR, a bile acid-activated nuclear receptor, has emerged as a promising strategy. FXR agonists like obeticholic acid exhibit time-dependent effects on glucose and lipid metabolism, mediated partly by gut microbiota-driven bile acid signaling [130]. These approaches could form the basis of next-generation chronobiome-informed therapeutics, provided they are validated in controlled human studies.

Future research should adopt a systems-level framework, integrating microbiome science, chrononutrition, endocrinology, and behavioral health. Machine learning techniques applied to multi-omics datasets could identify predictive signatures of circadian–microbiome misalignment and treatment responsiveness. Personalized chronotherapeutic platforms may eventually leverage an individual’s circadian phenotype, microbial configuration, and metabolic profile to guide interventions. In sum, while the interplay of circadian rhythms, social behavior, and the gut microbiome holds great promise for combating metabolic disease, clinical translation will require coordinated advances in mechanistic research, digital health, and personalized intervention design.

## 8. Conclusions

Social jetlag (SJL) represents a prevalent and quantifiable form of circadian misalignment with important implications for metabolic health. Emerging evidence highlights the gut microbiota as a key intermediary, linking SJL-induced desynchrony to inflammation, altered metabolite production, and impaired host metabolism. While mechanistic insights are often drawn from broader circadian disruption models, SJL-specific studies suggest comparable pathways and underscore its public health relevance. Interventions such as time-restricted eating, probiotics, and melatonin offer promising avenues for restoring circadian–microbial alignment, though targeted clinical studies remain limited. Addressing these research gaps will be essential to establish causality and to guide effective chronotherapeutic strategies.

## Figures and Tables

**Figure 1 medicina-61-01630-f001:**
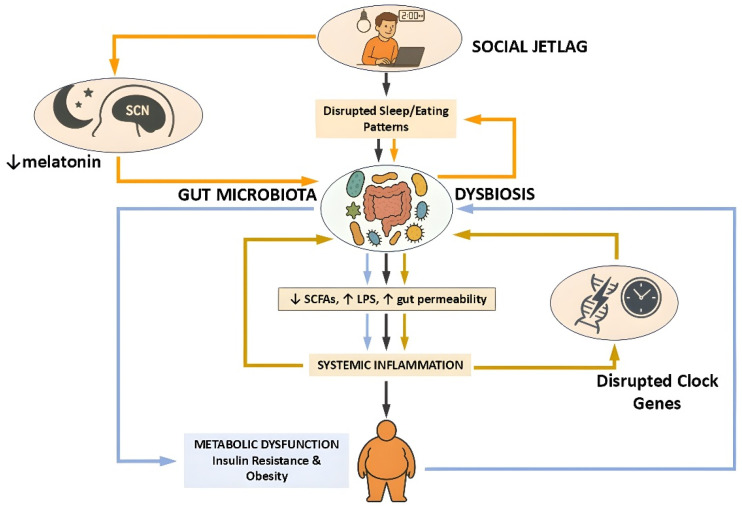
Schematic representation of the pathophysiological cascade linking social jetlag to metabolic dysfunction via the gut microbiota. Social jetlag, characterized by disrupted sleep and eating patterns, induces circadian misalignment that impairs gut microbial rhythmicity and composition. This dysbiosis is marked by decreased production of SCFAs, increased intestinal permeability, and elevated LPS translocation, collectively promoting systemic inflammation. The resulting immune activation contributes to insulin resistance and obesity. Additionally, suppression of melatonin and altered clock gene expression further exacerbate circadian and metabolic disturbances. Three interconnected feedback loops shape this pathophysiological network, each represented by color-coded arrows in the diagram. The orange loop highlights the central circadian pathway, in which social jetlag reduces melatonin secretion and disrupts the function of the suprachiasmatic nucleus (SCN), initiating microbial dysbiosis and subsequent metabolic impairment. The light blue loop traces the inflammatory–metabolic axis, where dysbiosis leads to systemic inflammation and metabolic dysfunction—processes that in turn can impair host circadian control and reinforce microbial instability. The golden-brown loop captures the reciprocal interaction between gut microbiota and peripheral circadian clock gene expression: microbial imbalance disrupts clock gene rhythmicity in peripheral tissues, which feeds back to destabilize microbial oscillations and inflammatory responses. This model emphasizes the bidirectional crosstalk among the gut microbiota, host circadian systems, and immune–metabolic regulation in the development of disease states associated with circadian disruption. Figure created using ChatGPT (OpenAI, GPT-4.5 model, https://chat.openai.com/ (accessed on 28 April 2025)) along with Microsoft PowerPoint (Microsoft Office Professional Plus 2021, Microsoft Corporation, Redmond, WA, USA).

**Table 1 medicina-61-01630-t001:** Chronotherapeutic strategies targeting the gut microbiota and their metabolic effects. Current chronotherapeutic interventions designed to modulate the gut microbiota in alignment with host circadian rhythms. Each strategy—ranging from probiotic and prebiotic administration to TRF, FMT, melatonin supplementation, and pharmacological modulation of bile acid pathways—targets microbial composition or function to restore metabolic homeostasis. Evidence from human and animal studies indicates that aligning these therapies with circadian timing enhances their effectiveness in improving glycemic control, reducing inflammation, and modulating gut-derived signaling. Abbreviations: PCOS, polycystic ovary syndrome; MetS, metabolic syndrome; FXR, farnesoid X receptor.

Intervention	Mechanism of Action	Observed Benefits
Timed Probiotic Supplementation	Aligns probiotic activity with host circadian rhythms.	Improved glycemic control and sleep quality in circadian-disrupted individuals.
Prebiotic Intake (e.g., *inulin*)	Enhances growth of beneficial microbes; synchronizes microbial metabolism.	Reduced insulin resistance and inflammation in women with PCOS.
Time-Restricted Feeding (TRF)/Time-Restricted Eating (TRE)	Restricts food intake to active phase; restores microbial rhythmicity and metabolic gene expression.	Improved insulin sensitivity and oxidative stress markers; protection against obesity in animal models.
Fecal Microbiota Transplantation (FMT)	Replaces dysbiotic microbiota; modifies microbial composition and function.	Increased *Lactobacillus* abundance; improved blood pressure in obese individuals with MetS.
Melatonin Supplementation	Synchronizes central and microbial clocks; modulates gut microbiota composition.	Decreased *Firmicutes*/*Bacteroidetes* ratio; reduced endotoxemia, inflammation, and insulin resistance.
FXR Agonists (e.g., *fexaramine*)	Targets bile acid signaling to influence gut–liver axis and microbial ecology.	Enhanced insulin sensitivity; increased adipose browning; circadian timing may augment efficacy.
Combined Chronotherapy Approaches	Leverages multiple synchronized interventions (e.g., TRF + probiotics) to enhance host–microbe circadian alignment.	Synergistic improvements in metabolic, inflammatory, and microbial parameters.

## Data Availability

Not applicable.

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
