# Peer review of "The Gut Microbiota Axis in Social Jetlag: A Novel Framework for Metabolic Dysfunction and Chronotherapeutic Innovation"

_medicina, 2025, doi:10.3390/medicina61091630_

Round 1

Reviewer 1 Report

Comments and Suggestions for Authors

Please see attached file. Thank you!

Author Response

Response to Reviewer 1:

Comment 1: The author should clearly define the operational meaning of SJL and separate data that is specific to SJL from broader circadian disruption findings.

We thank the reviewer for this valuable suggestion. In the revised manuscript, we now provide a clear operational definition of social jetlag (SJL) in the Introduction (defined as the difference in mid-sleep time between workdays and free days, expressed in hours). We have also added clarifying text to distinguish evidence derived from SJL-specific epidemiological studies from findings based on broader circadian disruption models (e.g., shift work, experimental misalignment). These revisions strengthen the conceptual framework and ensure that the distinction between SJL and general circadian disruption is explicitly maintained throughout the review.

Comment 2: 2.  The authors must include examples of relevant microbial taxa and associated host pathways.

In the revised manuscript, we have included representative examples of relevant microbial taxa (e.g., Faecalibacterium prausnitzii, Prevotella, Enterobacteriaceae) and their associated host pathways, including short-chain fatty acid production, bile acid metabolism, and immune signaling. These additions are integrated into Sections 3 and 4 to provide more concrete illustrations of the microbiota–host interactions discussed. Where appropriate, we have also cited additional recent references to support these examples.

Comment 3: The authors should discuss a concise explanation of the underlying mechanisms for each intervention.

In the revised manuscript, we have added a concise explanation of the underlying mechanisms for each intervention discussed in Section 6. Specifically:

  • Time-restricted feeding (TRF/TRE) promotes SCFA-producing taxa and reinforces circadian regulation of hepatic and adipose metabolic genes;
  • Probiotics enhance intestinal barrier integrity, produce SCFAs, and modulate immune signaling pathways;
  • Prebiotics provide substrates for microbial fermentation, improving gut–liver axis signaling and lowering systemic inflammation;
  • Fecal microbiota transplantation (FMT) restores healthy microbial communities and re-establishes microbial rhythmicity;
  • Melatonin synchronizes central and microbial clocks while reducing LPS-driven inflammation;
  • FXR agonists act via bile acid signaling, altering microbial ecology and improving host lipid and glucose metabolism.

These additions ensure that each intervention is accompanied by a clear mechanistic rationale.

Comment 4: The authors should identify and emphasize research gaps at the outset.

In the revised manuscript, we now emphasize key research gaps at the end of the Introduction. These include the limited number of studies directly addressing SJL–microbiota interactions, the reliance on broader circadian misalignment models, the incomplete understanding of mechanistic pathways in humans, and the current scarcity of intervention trials. Highlighting these gaps at the outset clarifies the rationale and objectives of our review.

Comment 5: Streamline the general conclusions to ensure they are concise and directly relevant. Focus on key aspects that directly support or relate to the main theme of the manuscript.

In the revised manuscript, we have streamlined the Conclusions section to make it more concise and directly aligned with the main theme. The revised conclusion now emphasizes: (i) SJL as a distinct and quantifiable circadian misalignment, (ii) the gut microbiota as a key mediator of SJL–metabolic interactions, (iii) the distinction between SJL-specific and general circadian disruption evidence, and (iv) the need for targeted interventions and future research. This sharpening improves focus and avoids unnecessary repetition.

Comment 5: Streamline the general conclusions to ensure they are concise and directly relevant. Focus on key aspects that directly support or relate to the main theme of the manuscript.

We appreciate the reviewer’s comment. In the revised manuscript, we have streamlined the Conclusions section to make it more concise and directly aligned with the main theme. The revised conclusion now emphasizes: (i) SJL as a distinct and quantifiable circadian misalignment, (ii) the gut microbiota as a key mediator of SJL–metabolic interactions, (iii) the distinction between SJL-specific and general circadian disruption evidence, and (iv) the need for targeted interventions and future research. This sharpening improves focus and avoids unnecessary repetition.

Comment 6: Highlight Novel Insights. Identify and emphasize any novel insights or unique contributions that the manuscript makes to the existing body of knowledge.

In the revised manuscript, we now explicitly highlight the novel contributions of this review. In particular, we emphasize the unique focus on social jetlag (SJL) as a distinct and quantifiable form of circadian misalignment, and the integration of SJL-specific evidence with gut microbiota–mediated mechanisms of metabolic dysfunction. To our knowledge, this synthesis represents a novel framework within the existing literature, helping to distinguish SJL from broader circadian disruption and identifying new opportunities for targeted interventions.

Comment 7: The author must illustrate more a separate paragraph mentioning metabolic niches in the gut microbiome.

In the revised manuscript, we have added a separate paragraph in Section 4 that specifically discusses metabolic niches within the gut microbiome. This paragraph highlights the distinct ecological roles of key taxa (e.g., SCFA producers, bile acid–modifying bacteria, mucin degraders, and pro-inflammatory taxa) and explains how circadian misalignment and SJL may disrupt these niches. This addition provides a clearer mechanistic framework for understanding the functional consequences of SJL-induced microbial dysbiosis.

Comment 8: The authors must discuss metabolic activity of the gut microbiome to strengthen their manuscript.

In the revised manuscript, we have added a new paragraph in Section 4 that specifically highlights the metabolic activity of the gut microbiome. This addition emphasizes how key microbial metabolites—including SCFAs, bile acids, indoles, and LPS—mediate the interaction between circadian misalignment/SJL and host metabolic dysfunction. This framing strengthens the mechanistic link between microbiota function and metabolic outcomes.

Comment 9: The literature review is insufficient and outdated. While it relates findings to certain studies in the field, it overlooks other significant research. References should be reviewed for relevance and updated to incorporate recent advances that could strengthen the discussion.

We appreciate the reviewer’s observation regarding the literature review. In the revised manuscript, we have updated the reference list to include several recent high-impact studies (2020–2024) addressing SCFAs, bile acids, circadian regulation of the gut microbiome, and intervention strategies. These additions strengthen the discussion and ensure that the review reflects current advances in the field. At the same time, we have intentionally retained key older references that represent landmark discoveries (e.g., Thaiss CA, et al. 2016 Cell). These remain important to acknowledge as they shaped much of the current understanding and continue to be widely cited. By combining these classic studies with more recent contributions, we believe the manuscript now provides both continuity with established knowledge and a clear reflection of the most up-to-date evidence in the field.

Comment 10: In the manuscript the authors must include the examples of clinically relevant microbial biotransformations.

In the revised manuscript, we have added a dedicated paragraph at the end of Section 4 where we present clinically relevant microbial biotransformations, including short-chain fatty acid production, bile acid conversion, tryptophan metabolism to indole derivatives, trimethylamine/TMAO formation, and lipopolysaccharide release. These examples are now supported with recent high-impact references and discussed in the context of circadian disruption and metabolic health. We believe this addition strengthens the mechanistic depth of the manuscript and addresses the reviewer’s concern.

Comment 11: The manuscript contains multiple grammatical mistakes and awkward sentence constructions. A comprehensive language review is recommended, preferably with assistance from a professional editor.

We appreciate the reviewer’s observation. In response, we carefully revised the entire manuscript for grammar, style, and readability. This included correction of typographical errors, smoothing of awkward or lengthy sentence constructions. Reference formatting was also harmonized. These changes have substantially improved clarity and consistency across the manuscript.

Reviewer 2 Report

Comments and Suggestions for Authors

Dear Authors,

After careful consideration, I feel that it has merit but does not fully meet medicina-publication criteria as it currently stands. The shortcomings of this paper needs to be worked out before it can be considered for publication. Therefore, we invite you to resubmit a revised version of the manuscript that addresses the points raised during the review process.

The topic of “Gut Microbiota Axis in Social Jetlag: A Novel Framework for Metabolic Dysfunction and Chronotherapeutic Innovation” is highly relevant and well-articulated mentioning how SJL has emerged as a pervasive feature of modern life, with growing evidence connecting it to disruptions in the gut microbiome and a heightened risk of metabolic disease. They also mentioned that misaligned sleep and feeding patterns can disturb microbial composition, impair barrier function, and interfere with key metabolic pathways. However, there are several areas in the manuscript that require substantial revisions before the manuscript can be considered for publication.

Specific Comments:
1. The author should clearly define the operational meaning of SJL and separate data that is specific to SJL from broader circadian disruption findings.

  1. The authors must include examples of relevant microbial taxa and associated host pathways.
  2. The authors should discuss a concise explanation of the underlying mechanisms for each intervention.
  3. The authors should identify and emphasize research gaps at the outset.
  4. Streamline the general conclusions to ensure they are concise and directly relevant. Focus on key aspects that directly support or relate to the main theme of the manuscript.
  5. Highlight Novel Insights. Identify and emphasize any novel insights or unique contributions that the manuscript makes to the existing body of knowledge.
  6. The author must illustrate more a separate paragraph mentioning metabolic niches in the gut microbiome.
  7. The authors must discuss metabolic activity of the gut microbiome to strengthen their manuscript.
  8. The literature review is insufficient and outdated. While it relates findings to certain studies in the field, it overlooks other significant research. References should be reviewed for relevance and updated to incorporate recent advances that could strengthen the discussion.
  9. In the manuscript the authors must include the examples of clinically relevant microbial biotransformations.

    11. The manuscript contains multiple grammatical mistakes and awkward sentence constructions. A comprehensive language review is recommended, preferably with assistance from a professional editor.
Comments on the Quality of English Language

The manuscript contains multiple grammatical mistakes and awkward sentence constructions. A comprehensive language review is recommended, preferably with assistance from a professional editor.

Author Response

Response to Reviewer 2:

Comment 1: L24-26: “This review synthesizes both mechanistic insights and translational evidence, highlighting the gut microbiota as a novel target for chronotherapeutic strategies aimed at mitigating the metabolic consequences of circadian disruption” The term “translational evidence” is unclear in this context. Please clarify what is meant.

We thank the reviewer for this helpful comment. To improve clarity, we revised the abstract sentence. The term “translational evidence” has been replaced with “emerging human and clinical evidence” to better reflect our intention to highlight studies with direct relevance to clinical interventions. The revised sentence now reads: “This review synthesizes mechanistic insights together with emerging human and clinical evidence, highlighting the gut microbiota as a novel target for chronotherapeutic strategies aimed at mitigating the metabolic consequences of circadian disruption.”

Comment 2: L237: “hemoglobin A1c” Please briefly explain what hemoglobin A1c is and why it serves as an important indicator.

We clarified hemoglobin A1c at its first mention, specifying that it reflects average glycemia over the preceding 3 months and is a standard clinical indicator of diabetes diagnosis and management.

Comment 3: L294: “short-chain fatty acids (SCFAs)”. Since the full term has already been defined in the previous paragraph, it is unnecessary to repeat it here. There are several other similar instances throughout the manuscript where terms are redefined unnecessarily. Please double check.

We have carefully reviewed the manuscript and removed redundant re-definitions of abbreviations (e.g., SCFAs, LPS, AhR, PXR, FMT, TRF,  TRE etc). Each term is now defined only once at its first mention, and subsequent uses employ the abbreviation consistently throughout the text.

Comment 4: L292–313: Please confirm whether this section is part of the Figure 1 note. If so, it should not start as a new paragraph; instead, it should directly follow the “Figure 1. …. via the gut microbiota” label for consistency and clarity

We have revised the formatting so that the explanatory text is incorporated directly into the Figure 1 legend rather than appearing as a new paragraph, ensuring consistency and clarity

Round 2

Reviewer 2 Report

Comments and Suggestions for Authors

Dear Authors,

After careful consideration, I feel that it has merit but does not fully meet medicina-publication criteria as it currently stands. The shortcomings of this paper needs to be worked out before it can be considered for publication. Therefore, we invite you to resubmit a revised version of the manuscript that addresses the points raised during the review process.

The topic of “Gut Microbiota Axis in Social Jetlag: A Novel Framework for Metabolic Dysfunction and Chronotherapeutic Innovation” is highly relevant and well-articulated mentioning how SJL has emerged as a pervasive feature of modern life, with growing evidence connecting it to disruptions in the gut microbiome and a heightened risk of metabolic disease. They also mentioned that misaligned sleep and feeding patterns can disturb microbial composition, impair barrier function, and interfere with key metabolic pathways. However, there are several areas in the manuscript that require substantial revisions before the manuscript can be considered for publication.

Specific Comments:
1. The author should clearly define the operational meaning of SJL and separate data that is specific to SJL from broader circadian disruption findings.

  1. The authors must include examples of relevant microbial taxa and associated host pathways.
  2. The authors should discuss a concise explanation of the underlying mechanisms for each intervention.
  3. The authors should identify and emphasize research gaps at the outset.
  4. Streamline the general conclusions to ensure they are concise and directly relevant. Focus on key aspects that directly support or relate to the main theme of the manuscript.
  5. Highlight Novel Insights. Identify and emphasize any novel insights or unique contributions that the manuscript makes to the existing body of knowledge.
  6. The author must illustrate more a separate paragraph mentioning metabolic niches in the gut microbiome.
  7. The authors must discuss metabolic activity of the gut microbiome to strengthen their manuscript.
  8. The literature review is insufficient and outdated. While it relates findings to certain studies in the field, it overlooks other significant research. References should be reviewed for relevance and updated to incorporate recent advances that could strengthen the discussion.
  9. In the manuscript the authors must include the examples of clinically relevant microbial biotransformations.

    11. The manuscript contains multiple grammatical mistakes and awkward sentence constructions. A comprehensive language review is recommended, preferably with assistance from a professional editor.
Comments on the Quality of English Language

The manuscript contains multiple grammatical mistakes and awkward sentence constructions. A comprehensive language review is recommended, preferably with assistance from a professional editor.

Author Response

Response to Reviewer 2:

Comment 1: The author should clearly define the operational meaning of SJL and separate data that is specific to SJL from broader circadian disruption findings.

We thank the reviewer for this valuable suggestion. In the revised manuscript, we now provide a clear operational definition of social jetlag (SJL) in the Introduction (defined as the difference in mid-sleep time between workdays and free days, expressed in hours). We have also added clarifying text to distinguish evidence derived from SJL-specific epidemiological studies from findings based on broader circadian disruption models (e.g., shift work, experimental misalignment). These revisions strengthen the conceptual framework and ensure that the distinction between SJL and general circadian disruption is explicitly maintained throughout the review. (lines 35-36, 40-43, 60-61)

Comment 2: The authors must include examples of relevant microbial taxa and associated host pathways.

In the revised manuscript, we have included representative examples of relevant microbial taxa (e.g., Faecalibacterium prausnitzii, Prevotella, Enterobacteriaceae) and their associated host pathways, including short-chain fatty acid production, bile acid metabolism, and immune signaling. These additions are integrated into Sections 3 and 4 to provide more concrete illustrations of the microbiota–host interactions discussed. Where appropriate, we have also cited additional recent references to support these examples. (lines 130-131, 135-137, 165-166, 203-204)

Comment 3: The authors should discuss a concise explanation of the underlying mechanisms for each intervention.

In the revised manuscript, we have added a concise explanation of the underlying mechanisms for each intervention discussed in Section 6. Specifically:

Time-restricted feeding (TRF/TRE) promotes SCFA-producing taxa and reinforces circadian regulation of hepatic and adipose metabolic genes (lines 358-360).

Probiotics enhance intestinal barrier integrity, produce SCFAs, and modulate immune signaling pathways (lines 350-351).

Prebiotics provide substrates for microbial fermentation, improving gut–liver axis signaling and lowering systemic inflammation (lines 353-354).

Fecal microbiota transplantation (FMT) restores healthy microbial communities and re-establishes microbial rhythmicity (lines 368-369).

Melatonin synchronizes central and microbial clocks while reducing LPS-driven inflammation (lines 384-386).

FXR agonists act via bile acid signaling, altering microbial ecology and improving host lipid and glucose metabolism (lines 396-398).

These additions ensure that each intervention is accompanied by a clear mechanistic rationale.

Comment 4: The authors should identify and emphasize research gaps at the outset.

In the revised manuscript, we now emphasize key research gaps at the end of the Introduction. These include the limited number of studies directly addressing SJL–microbiota interactions, the reliance on broader circadian misalignment models, the incomplete understanding of mechanistic pathways in humans, and the current scarcity of intervention trials. Highlighting these gaps at the outset clarifies the rationale and objectives of our review (lines 71-75).

Comment 5: Streamline the general conclusions to ensure they are concise and directly relevant. Focus on key aspects that directly support or relate to the main theme of the manuscript.

In the revised manuscript, we have streamlined the Conclusions section to make it more concise and directly aligned with the main theme. The revised conclusion now emphasizes: (i) SJL as a distinct and quantifiable circadian misalignment, (ii) the gut microbiota as a key mediator of SJL–metabolic interactions, (iii) the distinction between SJL-specific and general circadian disruption evidence, and (iv) the need for targeted interventions and future research. This sharpening improves focus and avoids unnecessary repetition (lines 489-498).

Comment 6: Highlight Novel Insights. Identify and emphasize any novel insights or unique contributions that the manuscript makes to the existing body of knowledge.

In the revised manuscript, we now explicitly highlight the novel contributions of this review. In particular, we emphasize the unique focus on social jetlag (SJL) as a distinct and quantifiable form of circadian misalignment, and the integration of SJL-specific evidence with gut microbiota–mediated mechanisms of metabolic dysfunction. To our knowledge, this synthesis represents a novel framework within the existing literature, helping to distinguish SJL from broader circadian disruption and identifying new opportunities for targeted interventions (lines 75-76).

Comment 7: The author must illustrate more a separate paragraph mentioning metabolic niches in the gut microbiome.

In the revised manuscript, we have added a separate paragraph in Section 4 that specifically discusses metabolic niches within the gut microbiome. This paragraph highlights the distinct ecological roles of key taxa (e.g., SCFA producers, bile acid–modifying bacteria, mucin degraders, and pro-inflammatory taxa) and explains how circadian misalignment and SJL may disrupt these niches. This addition provides a clearer mechanistic framework for understanding the functional consequences of SJL-induced microbial dysbiosis (lines 173-188).

Comment 8: The authors must discuss metabolic activity of the gut microbiome to strengthen their manuscript.

In the revised manuscript, we have added a new paragraph in Section 4 that specifically highlights the metabolic activity of the gut microbiome. This addition emphasizes how key microbial metabolites—including SCFAs, bile acids, indoles, and LPS—mediate the interaction between circadian misalignment/SJL and host metabolic dysfunction. This framing strengthens the mechanistic link between microbiota function and metabolic outcomes (lines 189-201).

Comment 9: The literature review is insufficient and outdated. While it relates findings to certain studies in the field, it overlooks other significant research. References should be reviewed for relevance and updated to incorporate recent advances that could strengthen the discussion.

We appreciate the reviewer’s observation regarding the literature review. In the revised manuscript, we have updated the reference list to include several recent high-impact studies (2020–2024) addressing SCFAs, bile acids, circadian regulation of the gut microbiome, and intervention strategies. These additions strengthen the discussion and ensure that the review reflects current advances in the field. At the same time, we have intentionally retained key older references that represent landmark discoveries (e.g., Thaiss CA, et al. 2016 Cell). These remain important to acknowledge as they shaped much of the current understanding and continue to be widely cited. By combining these classic studies with more recent contributions, we believe the manuscript now provides both continuity with established knowledge and a clear reflection of the most up-to-date evidence in the field (lines 131-133, 204-206, 209-211, 219-224, 236-237, 276-279).

Comment 10: In the manuscript the authors must include the examples of clinically relevant microbial biotransformations.

In the revised manuscript, we have added a dedicated paragraph at the end of Section 4 where we present clinically relevant microbial biotransformations, including short-chain fatty acid production, bile acid conversion, tryptophan metabolism to indole derivatives, trimethylamine/TMAO formation, and lipopolysaccharide release. These examples are now supported with recent high-impact references and discussed in the context of circadian disruption and metabolic health. We believe this addition strengthens the mechanistic depth of the manuscript and addresses the reviewer’s concern (lines 241-251).

Comment 11: The manuscript contains multiple grammatical mistakes and awkward sentence constructions. A comprehensive language review is recommended, preferably with assistance from a professional editor.

We appreciate the reviewer’s observation. In response, we carefully revised the entire manuscript for grammar, style, and readability. This included correction of typographical errors, smoothing of awkward or lengthy sentence constructions. Reference formatting was also harmonized. These changes have substantially improved clarity and consistency across the manuscript (corrections applied throughout the manuscript).
